# Fiscal Expenditure on Sports and Regional Carbon Emissions: Evidence from China

**Muwei Xi** [1], **Dingqing Wang** [2] and **Ye Xiang** [3],*

1   College of Physical Education, Beihua University, Jilin 132000, China
2   School of Economics, Jilin University, Changchun 130012, China
3   Physical Education College, Jilin University, Changchun 130012, China
*   Correspondence: xiangye@jlu.edu.cn

**Abstract:** Due to the problems of economic structure and the ways of industrial development, many countries have accumulated many ecological problems in the process of economic development, especially the increase in carbon emissions, the greenhouse effect, and the emergence of a series of problems, which makes the global ecosystem suffer severe challenges. Achieving green sustainable development has become a strategic development arrangement for all countries, and as the sports economy is closely linked to regional green development, it has become an important channel to influence regional green development by promoting sports development. Based on theoretical analysis, this paper empirically analyzes and tests the impact of government fiscal support for sports on regional carbon emissions and its mechanisms based on provincial panel data using baseline regressions, a mediated effects model, and a threshold model. We argue that the development of green and healthy sports can have a significant impact on regional green development and that government fiscal support for sports plays an important role in this process. We found that government fiscal support for sports can significantly reduce regional carbon emissions and thus promote regional green development. From the perspective of mechanism analysis, the government's fiscal support for sports leads to the orderly development of the sports industry and its associated industries, which improves the rationalization of the regional industrial structure and thus reduces regional carbon emissions. Moreover, as the carbon productivity of the region continues to rise, the impact of fiscal support for sports on regional low-carbon development gradually deepens. This paper confirms the important role of the government's emphasis on sports development in the process of achieving carbon peaking and carbon neutrality, verifies the mediating effect of industrial structure rationalization, and further quantifies the correlation through the threshold effect, extending the study of the influencing factors affecting regional carbon emissions.

**Keywords:** carbon emissions; fiscal expenditure on sports; industrial rationalization; threshold effect; green economy

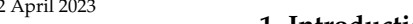

## 1. Introduction

While industrial civilization has created a huge material civilization for human society, it has also intensified the conflict between humans and nature. http://www.china.org.cn/chinese/2021-04/27/content_77445771.htm (accessed on 23 February 2023). The global ecological environment continues to deteriorate, natural crises caused by abnormal climate are becoming more frequent, and resources and energy supplies are becoming increasingly scarce [1]. Against this backdrop, the Paris Agreement was initiated by the United Nations and signed by some 178 parties worldwide, setting the development goal of "achieving a peak in global greenhouse gas emissions as soon as possible and net zero emissions by the second half of the century". It is clear that reducing carbon emissions and promoting a green economic transition are key to achieving sustainable regional development. In addition, the influence of the sports economy on economic and social development is

gradually increasing [2]. Research has shown that sports contribute to economic and social development, creating new jobs and stimulating infrastructure development [3,4]. It is clear that sport has become an important engine for global economic and social development [5], and there is a consensus on the economic and social effects of developing the sports industry. First, with the general public's quest for material well-being largely satisfied and the growing awareness of fitness and health, the sports industry has become a popular star industry around the world, which gives it enormous scope for development [6,7], and is gradually becoming an important part of the service sector [5]. Second, the development of sports brings many benefits to the economy and society; the sports industry not only creates new jobs and increases labor demand [3] but also stimulates infrastructure development [4] and accelerates urbanization [8]. In addition, the sports industry is increasingly linked to other industries [9], and the economic and social effects of sports continue to expand [10], which can contribute to the transformation of economic development and the upgrading of industrial structures [11]. However, with the increasing ecological pressure and the development of carbon neutrality and carbon peaking targets, the ecological effects of a green and healthy sports economy need to be considered. It is clear that reducing carbon emissions and promoting a green economic transition are key to achieving sustainable regional development [1] (Zhao et al., 2022). Therefore, it is important to explore the potential links that exist between fiscal expenditure on sports and regional carbon emissions. This is the focus of our attention.

The government is a guide and supporter of regional industrial development and plays an important role in the provision of public services. As a public health enterprise, the government plays an irreplaceable role in the development of sports. There is no denying that the government has made a significant contribution to the improvement of regional ecosystems through a range of economic initiatives, which can be divided into two main categories. On the one hand, there is direct intervention. The government has directly intervened in the allocation and operation of industries that have a significant impact on regional carbon emissions through quota systems [12], permit systems [13], approval systems [14], and direct public fiscal support [15], effectively preventing and controlling excessive regional pollutant emissions. On the other hand, there are indirect inducers. The main approaches are the establishment of an environmental tax [16], the creation of a carbon market [17], and a focus on increasing fiscal support for areas such as green innovation [18,19]. The government uses these measures to regulate the ecology of the region and induce businesses to achieve low-carbon production when it is profitable, thus effectively improving the ecology. However, governments have not paid enough attention to the ecological effects of sports business development [20,21], and the effects of the role of sports and health business on regional low-carbon development have been ignored by researchers. In recent years, the rapid development of sports has injected new impetus and vitality into the economy and society, and the concept of supporting the sports and health industry to promote regional low-carbon development has begun to receive attention [22]. Therefore, it is important to explore the emission reduction effects of fiscal support for sports to achieve sustainable development in the region.

Our study is based on provincial areas in China (we exclude Tibet, Hong Kong, Macao, and Taiwan based on the principle of data availability). The main reasons for choosing China as the subject of our study are as follows: On the one hand, China is the world's largest developing country, and it is in a critical period of economic transformation and comprehensive reform of its economic system. Under its long-term development strategy of prioritizing economic growth, China has accumulated many ecological and environmental problems [23,24] and economic system reform issues [25,26], and the goal of green development has been reached among all levels of government in the country. On the other hand, with China's GDP per capita already exceeding USD 10,000 by 2021, economic and social development has entered a new stage [27], residents are placing increasing emphasis on improving their lifestyles and ecological environment, high-quality development of sports that includes green economic attributes has emerged [28], and

society is placing increasing importance on the development of sports [29]. The Chinese government's Outline for the Construction of a Strong Sports Nation, released in 2019, proposes that the sports industry will become a pillar industry in China's national economy by 2035 and that the proportion of the sports industry in GDP will reach 4%, http://www.gov.cn/zhengce/2019-09/03/content_5426712.htm (accessed on 23 February 2023), which highlights the important role of the sports industry in economic development. Therefore, under the strategic objectives of pursuing "carbon neutrality" and "carbon peaking" and building a strong sports nation, the level of fiscal support for sports by local governments in China and its ecological effects deserve to be explored in depth and can provide lessons for other countries' fiscal systems and green economic development. It is also worth exploring the level of fiscal support for sports by local governments in China and its ecological effects.

Research in sports economics is relatively recent and an emerging subject area [30], and the abatement effects of fiscal support for sports are not common in existing studies. On the one hand, the literature has mainly used literature analysis [31–33], interviews and questionnaires [34,35], and simulation analysis [36,37] to focus on the study of the sports industry and the development analysis of the sports business itself, and the number of studies using relevant data to construct econometric models is relatively small. On the other hand, some scholars have explored the low-carbon development path of the sports industry itself [38–40], and the impact of the sports economy on regional low-carbon development [41], but none of them have considered the effect of governmental actions. Obviously, in promoting regional economic development, the government plays an important role in regional economic and social development through public finance spending [42,43], but there is a lack of studies exploring the effect of government announcement spending on regional low-carbon development from the perspective of the sports economy. Wynes (2021) and Ito and Higham (2023) also explored the carbon emission effects of sports business development only in terms of the shift in the way sports leagues play and the reduction in air travel [44,45]. We extend the previous literature on the emission reduction effects of fiscal expenditure on sports to verify the important impact of government actions on regional carbon emission levels in the context of sports public utility development from both theoretical and empirical perspectives. Specifically, our marginal contribution is mainly reflected in the following aspects: First, existing research on the sports industry has mainly focused on studying the development of the industry itself and the analysis of its economic and social effects. This paper extends the study of the ecological effects of sports business development by verifying the inhibiting effect of sports fiscal support on regional carbon emissions from the perspective of public fiscal support. Second, due to the difficulty of obtaining data on sports development, most existing studies have focused on questionnaires, simulations, and inductive analysis of literature, while few empirical studies have used econometric models. We obtain and use data related to the public expenditure on sports by provincial government departments in China to construct a provincial panel model of the relationship between government fiscal support and regional carbon emissions. Furthermore, we use carbon productivity as a threshold variable to verify the dynamic effect of government fiscal support on regional carbon emissions. Third, in the mechanism test, we investigate the transmission mechanism of government fiscal support for sports by optimizing the regional industrial structure and thus influencing regional carbon emissions from the perspective of industrial structure, which enriches the theoretical study of the ecological effects of fiscal support for sports by combining industrial economics theory.

Based on the theoretical analysis, this paper conducts an empirical analysis based on the collected panel data and uses OLS regression, mediated effects analysis, and threshold regression. The paper finds that government fiscal support for sports could significantly reduce the level of regional carbon emissions and thus promote regional green development. From the perspective of mechanism analysis, the government's fiscal support for sports could lead to the orderly development of the sports industry and its related industries, which improves the rationalization of regional industrial structure and thus reduces the level of regional carbon emissions. In addition, with the continuous improvement of

regional carbon productivity, the influence of fiscal support for sports undertakings on regional low-carbon development gradually deepens. The remainder of the paper is designed as follows: The second part is the theoretical analysis and hypotheses, where we provide a logical analysis of the paper's research themes and mechanisms. The third part is the research design, where we introduce the analytical model involved in this paper and explain the relevant variables. The fourth part is the empirical analysis and testing. The fifth part is the conclusion and implications.

## 2. Theoretical Analysis and Research Hypothesis

As countries around the world move toward the goal of sustainable development, low-carbon goals to drive the low-carbon and green development of the sports industry, through technological innovation, industrial transformation, new energy, and other means to pull the green low-carbon sports market and its sports consumption transformation [46], are better to achieve a win–win situation for sports, economic and social development, and ecological environmental protection [47]. The flourishing development of sports is closely linked to the green development of the region, and green sports and healthy sports economies have a profound impact on the level of green development of the region [48,49]. As a guide for sports development and a promoter of sustainable development in the region, the Chinese government will guide and support the development of sports through fiscal expenditure, giving full play to the ecological effects of sports development [2], which will reduce regional carbon emissions and thus enhance the level of regional green development.

From the supply side, increasing fiscal support for sports can, on the one hand, develop smart manufacturing, energy saving, and emission-reducing production, select low-carbon material fabrics [50], integrate green and low-carbon concepts into the whole process of production, transportation, and sales of sports manufacturing enterprises, and produce low-carbon sports clothing, shoes, hats, and equipment, thus effectively reducing regional carbon emissions. On the other hand, the development of a low-carbon fitness and leisure industry can be explored by setting relevant standards, cultivating typical cases, and promoting pilot projects to create low-carbon fitness and leisure demonstration zones and new models [51], which will help promote the reduction of regional carbon emissions. From the demand side, the government, by financially supporting the organization of large-scale sporting events, guides the public to change their lifestyles, places emphasis on cultivating residents' sporting health and wellness habits, and advocates low-carbon living, thus promoting regional green development [52]. The bidding concept for the 2022 Beijing Winter Olympics is "athlete-centered, sustainable and frugal", with the concept of "sustainable development" emphasizing "development that meets the needs of the present without compromising the future needs of the present generation". The concept of "sustainable development" emphasizes "development that meets the needs of today's people without compromising their future needs and gives full play to the promotion and regulation of the Olympic Movement on the economy, society, and the natural environment.", https://new. inews.gtimg.com/tnews/5757e2c6/9fec/5757e2c6-9fec-41a0-9a96-3ad963594bca.pdf (accessed on 23 February 2023), which aims to integrate the concept of green development throughout preparatory work. The green development concept is closely linked to the low-carbon lifestyle of residents through the development of sport, gradually transforming their low-carbon lifestyle and promoting the development of an ecological society, forming a virtuous cycle of green ecological environment and economic and social development. Therefore, we propose the following hypothesis.

**Hypothesis 1.** *Fiscal support for sports can reduce regional carbon emissions, which can contribute to regional green development.*

Rationalization of industrial structure enhances the proportional balance and coordination between industries through the reallocation of factor resources between different

industrial sectors. The rationalization and improvement of the industrial structure guarantee the proportional and coordinated development of the national economy and thus efficient and sustainable development in a virtuous circle [53]. Fiscal support for sports can effectively improve the degree of rationalization of the regional industrial structure. The industrial structure of a region determines its industrial ecology, which has a strong influence on the green development of the region from both the supply and demand sides of the industrial chain. On the one hand, under the goal of "carbon peaking" and "carbon neutrality", the carbon emissions of sports enterprises will become a market price signal for changes in the structure of the sports industry, and the sports carbon emissions trading market will become an important factor in promoting the low-carbon development of sports enterprises. Fiscal support for sports can facilitate the rational allocation of sports resources and promote reasonable changes in sports industries. Sports enterprises with high energy consumption and high emissions will pay more to develop low-carbon production, while sports enterprises with low energy consumption and low pollution will have greater market competitiveness, and the rationalization of the industry will be enhanced, thus enabling the regional carbon emissions to be effectively controlled. On the other hand, the sports convergence industry is currently emerging in China, and the orderly development of sports can promote the flourishing of related services, product processing industries, and some advanced technologies [54]. Under the leadership of low-carbon living, residents' consumption structure is becoming more rational, and their demand for the quality of sports consumption is gradually increasing, prompting the sports industry to respond quickly to the needs of residents and facilitating the rationalization of the industry. From the perspective of industrial ecology theory, the rationalization of industrial structure can provide an impetus for the development of industrial ecology [55], which in turn reduces regional carbon emissions [56]. Therefore, we propose the following hypothesis.

**Hypothesis 2.** *Fiscal support for sports can reduce regional carbon emissions by increasing the rationalization of industries.*

The steady increase in carbon productivity is a manifestation of an optimal transformation of the economic development approach, technological progress, and a significant increase in resource efficiency [57]. First, the increase in carbon productivity indicates a greener and more sustainable approach to economic development. The green and health-oriented sports industry is in line with the theme of sustainable development, and the industrial chain associated with sports is gradually formed under the goal of sustainable development, which makes the contribution of government-supported and government-guided sports to economic and social development increase, and ultimately, the ecological effect of government fiscal support for sports will be significantly improved. Second, the increase in carbon productivity may be the result of green technological innovation and technological progress. Under the original production model, technological progress can significantly improve the input–output ratio, which makes the green economy effect of government fiscal expenditure on sports and health more obvious and reduces carbon emissions more significantly. Third, the increase in carbon productivity may be the result of a significant increase in resource efficiency. The ecological effects of the development of a green and health-oriented sports industry are further amplified, which makes the effect of government fiscal support for sports on the reduction of carbon emissions per unit more obvious and thus more conducive to the green and sustainable development of the region. In summary, we propose the following hypothesis.

**Hypothesis 3.** *The effect of fiscal support for sports on regional carbon emissions has a threshold effect based on carbon productivity. That is, as carbon productivity increases, the more significant the emission reduction effect of fiscal support for sports is.*

## 3. Research Design

### 3.1. Regression Model

To test the effect of fiscal support for sports on regional carbon emissions, that is, Hypothesis 1, we construct the basic analytical model of this paper by drawing on the model used by Wang et al. (2022) [2] to study a similar problem.

$$carbon\_emi_{it} = \beta_0 + \beta_1 sports_{it} + \beta_2 r\&d_{it} + \beta_3 energy_{it} + \beta_4 patent_{it} + \beta_5 f di_{it} + \beta_6 g dp_{it} + \varepsilon_{it} \tag{1}$$

Here, *carbon_emi* represents regional carbon emissions, and we use regional carbon emissions per capita to measure the level of regional carbon emissions. *sports* represents regional fiscal support for sports and is expressed in terms of the stock of fiscal expenditure on sports. *energy* is the amount of energy consumed per unit of GDP and represents the level of energy consumption. *r&d* is the rate of change in investment in research and development, representing the degree of importance attached to regional scientific and technological development. *patent* is the number of invention patents granted per 10,000 people, representing the foundation and atmosphere of regional innovation. *fdi* is the amount of actual foreign direct investment per capita, which represents the opening level of the region to the outside world. *gdp* is the regional GDP per capita, representing the level of regional economic development. $\beta$ is the regression coefficient, and $\varepsilon$ is the error term.

To verify Hypothesis 2, we explore the transmission path of fiscal support for sports through changing government expenditure preferences on healthcare and social security and employment and thus affecting regional carbon emissions from the perspective of livelihood-based expenditure. This paper draws on the design of the mediation model by Baron and Kenny [58] and uses the stepwise regression method to test the mediation effect, where *rationalization_ind* is the degree of industrial rationalization. The diagram of the transmission mechanism is shown in Figure 1.

$$rationalization\_ind_{it} = \gamma_0 + \gamma_1 sports_{it} + \sum_{j=2}^{6} \gamma_j control_{jit} + \varepsilon_{it} \tag{2}$$

$$carbon\_emi_{it} = \varphi_0 + \varphi_1 sports_{it} + \varphi_2 rationalization\_ind + \sum_{j=3}^{7} \varphi_j control_{jit} + \varepsilon_{it} \tag{3}$$

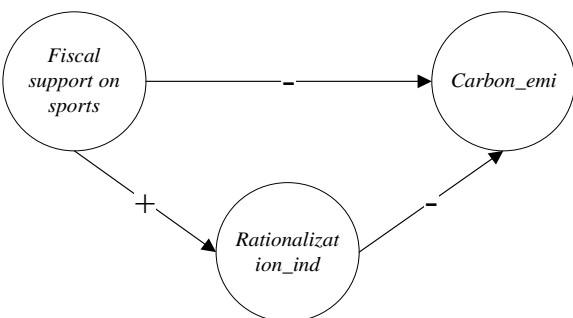

**Figure 1.** The mechanism of transmission.

To examine the non-linear effects of carbon productivity on carbon emissions, the following single-, double-, and triple-threshold regression models were constructed using Hansen's [59] panel threshold regression model. where carbon productivity (*carbon_pro*) is the threshold variable, $\gamma$ is the threshold to be estimated, *I* is the indicator function, and $X_{it}$ is the sport fiscal expenditure and control variables.

$$carbon\_emi_{it} = \varphi_0 + \varphi_1 sports_{it} \cdot I(carbon\_pro \leq \gamma) + \varphi_2 sports_{it} \cdot I(carbon\_pro > \gamma) + \varphi X_{it} + \varepsilon_{it} \tag{4}$$

$$carbon\_emi_{it} = \varphi_0 + \varphi_1 sports_{it} \cdot I(carbon\_pro \leq \gamma_1) + \varphi_2 sports_{it} \cdot I(\gamma_1 < carbon\_pro \leq \gamma_2)$$
$$+ \varphi_3 sports_{it} \cdot I(carbon\_pro > \gamma_2) + \varphi X_{it} + \varepsilon_{it} \tag{5}$$

$$
\begin{aligned}
carbon\_emi_{it} &= \varphi_0 + \varphi_1 sports_{it} \cdot I(carbon\_pro \leq \gamma_1) + \varphi_2 sports_{it} \cdot I(\gamma_1 < carbon\_pro \leq \gamma_2) \\
&+ \varphi_3 sports_{it} \cdot I(\gamma_2 < carbon\_pro \leq \gamma_3) + \varphi_4 sports_{it} \cdot I(carbon\_pro > \gamma_3) + \varphi X_{it} + \varepsilon_{it}
\end{aligned}
\tag{6}
$$

### 3.2. *Variable Description*

#### 3.2.1. Core Explanatory Variable

We used data on public expenditure on sports by region from the China Sports Statistics Yearbook as a measure of the level of public expenditure on sports. In the data processing, we took 2006 as the initial year, calculated the stock of public expenditure on sports according to the perpetual inventory method, and took logarithms.

#### 3.2.2. Mediating Variable

The rationalization of industrial structure is expressed as the degree of coordination between industries and the degree of effective use of resources, and this paper uses the Thiel index as its proxy variable. The calculation formula is as follows [40]:

$$
rationalization\_ind_{it} = 1/TL = 1 \Big/ \sum\nolimits_{i=1}^{n} \frac{Y_i}{Y} \ln\left( \frac{Y_i}{L_i} \Big/ \frac{Y}{L} \right)
\tag{7}
$$

where *rationalization_ind* is the degree of industrial rationalization. *TL* is the Thiel index, which indicates the degree of deviation of the industrial structure from the equilibrium state. *Y* indicates the output value. *L* indicates the number of employees. $i = 1, 2,$ and 3, representing the primary, secondary, and tertiary industries, respectively. In addition, the larger the value of the Thiel index is, the lower the degree of industrial structure rationalization, and the smaller the value of the Thiel index is, the higher the degree of industrial structure rationalization.

#### 3.2.3. Threshold Variable

Carbon productivity (*carbon_pro*) is the GDP generated per unit of $CO_2$ emissions and is the ratio of regional GDP to $CO_2$ emissions, which is the reciprocal of carbon intensity. A higher value of carbon productivity means more efficient economic growth in the sense that more output is produced with less energy consumption.

#### 3.2.4. Control Variables

The control variables in this paper mainly include the level of energy consumption (*energy*), R&D support (*r&d*), innovation base and climate (*patent*), openness to the outside world (*fdi*), and per capita gross regional product (*gdp*). First, the level of energy consumption (*energy*) represents the level of energy consumption, which may influence regional carbon emissions through a variety of channels. Second, the regional R&D investment can change the level of regional green innovation, which in turn may affect the level of regional carbon emissions [60]; therefore, we introduce R&D support (*r&d*) and innovation base and climate (*patent*) as control variables in the model. Third, some studies also suggest that trade openness may contribute to the development of the renewable energy industry [61], and the introduction of external capital may also promote the development of the energy industry and thus affect the level of regional carbon emissions, so we introduce the level of external openness (*fdi*) as a control variable into the model. Fourth, we consider gross regional product (*gdp*) per capita as a non-negligible control variable. We argue that as the income level of regional residents increases, the diversified consumption of residents and the use of high energy-consuming equipment increase regional carbon emissions. Of course, as the income level of regional residents and technological progress continue to increase, residents' consumption may be tilted toward green development. Therefore, the effect of regional gross domestic product (*gdp*) per capita on regional carbon emissions may be non-linear, and we add the primary and secondary terms of this variable to the model equation. The descriptive statistics of the variables are shown in Table 1.

**Table 1.** Descriptive statistics for variables.

| VarName | Obs | Mean | Median | SD | Min | Max | Unit |
|---|---|---|---|---|---|---|---|
| *carbon_emi* | 360 | 9.643 | 7.657 | 6.310 | 2.387 | 37.602 | ton/person |
| *sports* | 360 | 12.901 | 13.000 | 0.828 | 9.717 | 14.527 | ln(CNY 10,000) |
| *rationalization_ind* | 360 | 7.493 | 4.498 | 9.034 | 1.173 | 57.266 | \ |
| *carbon_ pro* | 360 | 2.911 | 2.341 | 1.975 | 0.403 | 12.754 | ton/CNY 10,000 |
| *gdp* | 360 | 4.059 | 3.569 | 2.408 | 0.579 | 12.899 | CNY 10,000/person |
| *energy* | 360 | 1.089 | 0.934 | 0.617 | 0.239 | 4.142 | ton/CNY 10,000 |
| *fdi* | 360 | 0.108 | 0.069 | 0.133 | 0.001 | 0.851 | CNY 10,000 |
| *r&d* | 360 | 2.186 | −0.196 | 7.706 | −0.946 | 66.562 | % |
| *patent* | 360 | 1.110 | 0.405 | 2.239 | 0.036 | 21.230 | Numbers/10,000 people |

*3.3. Data Sources*

We explore the impact of government sports fiscal spending on regional carbon emissions based on provincial panel data from 2006 to 2017 in China. There are several reasons for choosing this interval for our data. On the one hand, the statistical caliber of fiscal expenditure in China was changed in 2006, and we set the starting year as 2006 to maintain the consistency of the statistical caliber of fiscal expenditure-related data. On the other hand, under the principle of data availability, we obtained data on sports expenditures up to 2017. In addition, there are four main sources of data we used. The first is data on government fiscal expenditure on sports, with the original data coming from the China Sports Statistical Yearbook (These data are not publicly disclosed on the official website, so please contact the manuscript author if you have any questions.). The second is data on regional carbon emission levels, with the original data sourced from the China Energy Statistical Yearbook, https://data.stats.gov.cn/easyquery.htm?cn=C01 (accessed on 21 March 2023). The third is the data involved in this study, including mediating variables, threshold variables, and control variables, with the original data mainly sourced from the National Bureau of Statistics, https://data.stats.gov.cn/ (accessed on 21 March 2023).

**4. Regression Analysis**

*4.1. Baseline Regression Analysis*

We estimated Equation (1) using a fixed effects model, and the regression results are shown in model 1 and model 2 of Table 2. We find that the regression coefficients of fiscal expenditure on sports on regional carbon emissions are both significantly negative at the 1% confidence interval. To test the robustness of the regression results, we adopted a regression analysis by replacing the regression model. Model 3 is regressed using the truncated regression model (There is left censoring in the data of the dependent variable. Therefore, we chose to use the truncated regression model for testing.) Models 4 and 5 were regressed using a random effects model and Tobit model, respectively. The estimation results are generally consistent, which indicates that fiscal support for sports can significantly reduce the level of regional carbon emissions.

From the regression results of the control variables, the regression coefficients of the primary and secondary terms of per capita gross regional product (*gdp*) on regional carbon emissions are one positive and one negative, respectively, which indicates that regional carbon emissions increase and then decrease as the level of regional development increases. From the results of Model 2 after controlling for two-way fixed effects, the inflection point of the effect of GDP per capita is about CNY 120,600. This indicates that in the process of continuous economic development, regions pay more attention to the sustainability of economic development. The regression coefficients of regional energy consumption levels (*energy*) on regional carbon emissions are all negative and pass the significance test at the 1% confidence interval. This result may seem surprising, but in practical terms, such an empirical analysis is reasonable. The decreasing amount of energy consumption per unit of GDP due to technological progress has been accompanied by increasing living standards and consumption levels of the population, which has increased

per capita carbon emissions. The regression coefficients of innovation base and atmosphere (*patent*) on regional carbon emissions are all negative and pass the significance test at the 5% confidence level or above, effectively verifying the importance of building an innovative environment for regional low-carbon development. The regression coefficients of external openness (*fdi*) on regional carbon emissions were negative but did not pass the significance test, indicating that effectively attracting foreign investment is an important measure for low-carbon development, and attention should be given to the directional guidance of foreign investment. The coefficient of R&D support (*r&d*) on regional carbon emissions is positive after controlling for individual effects, but this effect is not significant.

**Table 2.** Baseline regression result.

|  | (1) | (2) | (3) | (4) | (5) |
|---|---|---|---|---|---|
|  | carbon_emi | carbon_emi | carbon_emi | carbon_emi | carbon_emi |
| *sports* | −1.821 *** | −1.847 *** | −1.426 *** | −1.824 *** | −1.849 *** |
|  | (0.570) | (0.604) | (0.408) | (0.543) | (0.541) |
| *gdp2* | −0.075 *** | −0.089 *** | −0.244 *** | −0.110 *** | −0.088 *** |
|  | (0.022) | (0.029) | (0.035) | (0.023) | (0.022) |
| *gdp* | 1.764 *** | 2.147 *** | 4.968 *** | 2.544 *** | 2.053 *** |
|  | (0.314) | (0.505) | (0.348) | (0.309) | (0.311) |
| *energy* | −2.699 *** | −3.180 *** | −3.947 *** | −1.823 *** | −1.896 *** |
|  | (0.636) | (0.713) | (0.520) | (0.620) | (0.646) |
| *patent* | −0.322 ** | −0.219 ** | −0.440 ** | −0.427 *** | −0.371 ** |
|  | (0.161) | (0.095) | (0.189) | (0.152) | (0.152) |
| *fdi* | −1.710 | −2.644 | −4.947 ** | −2.697 | −2.056 |
|  | (2.182) | (2.254) | (2.514) | (2.251) | (2.133) |
| *r&d* | 0.032 | 0.058 | 0.048 | 0.016 | 0.010 |
|  | (0.052) | (0.055) | (0.032) | (0.045) | (0.048) |
| *constant* | 31.063 *** | 30.592 *** | 7.137 * | 26.622 *** | 29.793 *** |
|  | (7.137) | (7.871) | (4.326) | (6.877) | (6.894) |
| Individual fixed | YES | YES |  |  |  |
| Time fixed |  | YES |  |  |  |
| Wald test |  |  | 635.30 *** | 243.90 *** | 287.36 *** |
| adj. $R^2$ | 0.920 | 0.919 |  |  |  |
| N | 360 | 360 | 360 | 360 |  |

The values in brackets are SDs. *** Indicates that the estimated coefficients are significant at the confidence level of 1%. ** Indicates that the estimated coefficients are significant at the confidence level of 5%. * Indicates that the estimated coefficients are significant at the confidence level of 10%. The below is the same.

To circumvent possible endogeneity problems between variables and to refer to common methods used in existing studies, this paper uses one-period lagged fiscal support for sport (*L. sports*) as an instrumental variable. The one-period lag of the variable is correlated with the current period of the variable, but the lagged variable is not correlated with the current period disturbance term, so the one-period lag of fiscal support for sport can be a valid instrumental variable. In this paper, the two-stage least squares (2SLS), optimal GMM, and iterative GMM methods are used to address the potential endogeneity of the model, and the results of the analysis are shown in Table 3. The shea's adj. partial $R^2$ value is 0.7300, the F test statistic is 119.971, the *p*-value is 0.000, and the minimum eigenvalue statistic of 944.319 is much greater than 10, indicating that there are no weak instrumental variables. The results of the two-stage least squares (2SLS), optimal GMM, and iterative GMM estimations from model 1 to model 3 show that the regression coefficient of sports fiscal support on regional carbon emissions is significantly negative at the 1% confidence level, the results of the optimal GMM estimation and iterative GMM estimation are consistent with the IV-2SLS estimation results, and the enhancement of sports fiscal support can significantly reduce regional carbon emissions. The results of the regression coefficients of the optimal GMM estimation and the iterative GMM estimation are generally consistent with the results of IV-2SLS. Hypothesis 1 was verified.

**Table 3.** Endogeneity test.

|  | (1) | (2) | (3) |
|---|---|---|---|
|  | *carbon_emi* | *carbon_emi* | *carbon_emi* |
| *sports* | −3.138 *** | −3.138 *** | −3.138 *** |
|  | (0.888) | (0.888) | (0.888) |
| *gdp2* | −0.078 *** | −0.078 *** | −0.078 *** |
|  | (0.026) | (0.026) | (0.026) |
| *gdp* | 1.865 *** | 1.865 *** | 1.865 *** |
|  | (0.541) | (0.541) | (0.541) |
| *energy* | −2.581 | −2.581 | −2.581 |
|  | (1.667) | (1.667) | (1.667) |
| *patent* | −0.254 * | −0.254 * | −0.254 * |
|  | (0.140) | (0.140) | (0.140) |
| *fdi* | −1.853 | −1.853 | −1.853 |
|  | (1.599) | (1.599) | (1.599) |
| *r&d* | 0.060 * | 0.060 * | 0.060 * |
|  | (0.033) | (0.033) | (0.033) |
| *constant* | 42.020 *** | 42.020 *** | 42.020 *** |
|  | (13.250) | (13.250) | (13.250) |
| Individual fixed | YES | YES | YES |
| Time fixed | YES | YES | YES |
| Shea's adj. partial $R^2$ | 0.7300 |  |  |
| Robust F | 119.971 |  |  |
|  | [0.000] |  |  |
| Minimum eigenvalue statistic | 944.319 |  |  |
| adj. $R^2$ | 0.928 | 0.928 | 0.928 |
| N | 330 | 330 | 330 |

The values in brackets "( )" are SDs. The values in brackets "[ ]" are the value of p of the corresponding test statistics. The values in brackets are SDs. *** Indicates that the estimated coefficients are significant at the confidence level of 1%. ** Indicates that the estimated coefficients are significant at the confidence level of 5%. * Indicates that the estimated coefficients are significant at the confidence level of 10%.

### 4.2. Analysis of the Transmission Mechanism

From the previous analysis, it is clear that fiscal support for sports has a significant inhibiting effect on regional carbon emission levels. However, whether government-led sports economic development can promote the rationalization of industrial structure and thus reduce regional carbon emissions needs further verification. We adopted a stepwise analysis of mediating effects to regress Equation (3) on Equation (4), and the results are shown in Table 4. The results of the regression of the mediating effect on the degree of industrial rationalization show that the regression coefficient of fiscal support for sports on the degree of industrial rationalization is significantly positive at the 1% confidence level. This means that by providing fiscal support to the sports industry, the government promoted the rationalization of the industrial structure, which in turn suppressed the increase in regional carbon emissions. The mediating effect was −0.148 (2.740 × −0.054). From the results of the mediating effect test, it can be seen that the results of the Sobel test and the bootstrap test are consistent and hold at 5% confidence levels, and Hypothesis 2 is verified. An increase in government fiscal support for sports can enhance the rationalization of the regional industrial structure, which in turn suppresses the growth of regional carbon emission levels.

### 4.3. Threshold Effect Test of Carbon Productivity

To further investigate the effect of government fiscal support on regional carbon emissions at different levels of carbon productivity and to examine the threshold effect of regional carbon productivity, we drew on the threshold model constructed by Hansen (1999) [59] and conducted a regression analysis with regional carbon productivity as the threshold variable. From the results of the self-sampling test of the threshold effect and the estimation of the threshold value in Table 5, it can be seen that there is a

double-threshold effect on the influence of regional carbon productivity on the level of regional carbon emissions. The regression results of the effect of fiscal support for sports on regional carbon emission levels under different carbon productivity conditions are shown in Table 6. The regression coefficient of sports fiscal support on regional carbon emissions is −1.478 when carbon productivity is below 0.3675, the regression coefficient of sports fiscal support on regional carbon emissions is −1.658 when carbon productivity is in the range of 0.3675 to 0.6443, and the regression coefficient of sports fiscal support on regional carbon emissions is −1.892 when carbon productivity is above 0.6443. Overall, the absolute value of the regression coefficient of sports fiscal support on regional carbon emissions becomes larger and more significant as regional carbon productivity continues to increase. This indicates that the impact of the sports economy on regional low-carbon development deepens as regional carbon productivity increases and that fiscal support for sports has a non-linear impact on regional carbon emissions. Hypothesis 3 was verified.

**Table 4.** The regression result of the mediating effect based on the degree of industrial rationalization.

| | (1) | (2) | (3) |
|---|---|---|---|
| | *carbon_emi* | *rationalization_ind* | *carbon_emi* |
| *sports* | −1.847 *** | 2.740 *** | −1.699 *** |
| | (0.604) | (1.046) | (0.609) |
| *rationalization_ind* | | | −0.054 * |
| | | | (0.033) |
| *gdp2* | −0.089 *** | 0.246 *** | −0.075 ** |
| | (0.029) | (0.049) | (0.030) |
| *gdp* | 2.147 *** | −3.895 *** | 1.937 *** |
| | (0.505) | (0.875) | (0.520) |
| *energy* | −3.180 *** | 3.845 *** | −2.973 *** |
| | (0.713) | (1.234) | (0.722) |
| *patent* | −3.180 *** | 3.845 *** | −2.973 *** |
| | (0.713) | (1.234) | (0.722) |
| *fdi* | −0.219 | 0.153 | −0.211 |
| | (0.178) | (0.307) | (0.177) |
| *r&d* | −2.644 | -2.289 | −2.767 |
| | (2.254) | (3.901) | (2.249) |
| *constant* | 0.058 | 0.119 | 0.065 |
| | (0.055) | (0.096) | (0.055) |
| adj. $R^2$ | 0.919 | 0.882 | 0.920 |
| N | 360 | 360 | 360 |
| Sobel Test | | −0.148 * | |
| | | ($p = 0.062$ z $= −1.862$) | |
| Bootstrap Test | | −0.148 ** | |
| | | ($p = 0.034$ z $= −2.12$) | |

The values in brackets are SDs. *** Indicates that the estimated coefficients are significant at the confidence level of 1%. ** Indicates that the estimated coefficients are significant at the confidence level of 5%. * Indicates that the estimated coefficients are significant at the confidence level of 10%.

**Table 5.** Existence test of the threshold effect of regional carbon productivity.

| Threshold Variables | Threshold Type | F Value | *p* Value | Threshold Value | BS Times |
|---|---|---|---|---|---|
| *carbon_pro* | Single threshold | 87.20 | 0.0133 | 0.6024 | 300 |
| | Double threshold | 54.12 | 0.0133 | 0.3675 0.6443 | |
| | Triple threshold | 50.13 | 0.5833 | | |

**Table 6.** Threshold regression results.

|  | (Double Threshold) |
|---|---|
|  | *carbon_emi* |
| *sports* (*carbon_ pro* ≤ 0.3675) | −1.478 *** |
|  | (0.481) |
| *sports* (0.3675 < *carbon_ pro* ≤ 0.6443) | −1.658 *** |
|  | (0.480) |
| *sports* (*carbon_ pro* > 0.6443) | −1.892 *** |
|  | (0.480) |
| *constant* | 24.115 *** |
|  | (6.043) |
| *control* | YES |
| adj. $R^2$ | 0.582 |
| N | 360 |

The values in brackets are SDs. *** Indicates that the estimated coefficients are significant at the confidence level of 1%. ** Indicates that the estimated coefficients are significant at the confidence level of 5%. * Indicates that the estimated coefficients are significant at the confidence level of 10%.

## 5. Conclusions and Implications

As a guide and fiscal supporter in times of economic transition, the government's fiscal support behavior plays a pivotal role in the process of regional low-carbon development. Unlike previous studies that focus on the economic effects of government fiscal support, we focus more on the impact of fiscal support for sports on regional carbon emissions and its mechanisms. We answer the question of whether fiscal support for sports reduces regional carbon emissions based on provincial panel data in China. Our study finds the following conclusions: First, government fiscal support for sports can significantly reduce regional carbon emissions. Second, fiscal support for sports can significantly reduce regional carbon emissions by increasing the rationalization of industries, thus promoting green development in the region. Third, the emission reduction effect of fiscal support for sports becomes increasingly significant as carbon productivity increases; i.e., carbon productivity has a threshold effect on the impact of fiscal support for sports on regional carbon emissions. Therefore, our study has implications for the development of sports and the promotion of the green economy in transition economies to better achieve the goal of carbon peaking and carbon neutrality.

Based on the results of our study, we have the following policy insights: First, there is a need for the government to recognize the economic potential and value of sports, and invest in sports infrastructure development appropriate to the stage of economic development in the region. The government should also promote the green concept in sports development to enhance ecological and environmental awareness, reduce energy consumption levels, and promote regional green development through green sports. This insight reflects the logic that sports can be a powerful tool for promoting regional economic growth and environmental sustainability. Second, we should focus on the mediating effect of industrial structure rationalization on regional carbon emissions. The government should guide social capital into the sports and health industry through fiscal support and promote the flourishing of green services in the form of industrial support, while optimizing the regional industrial structure. Regions should make full use of the opportunity of industrial gradient transfer to encourage the application of new technologies to transform the traditional sports industry sector and strive to improve the degree of industrial rationalization, thereby accelerating the green development of the region. Third, government departments should fully recognize that a sustained increase in regional carbon productivity can more effectively leverage the emission reduction effect of fiscal support for sports. The government should increase its support for low-carbon economic development policies, advocate technological innovation in enterprises, improve the efficiency of energy resource use, and increase the output value per unit of energy consumption so that the emission reduction effect of fiscal support for sports can be given full play. Overall, the passage provides valuable policy

insights based on empirical evidence, highlighting the important role of sports in promoting regional economic growth and environmental sustainability. The insights are grounded in a logical framework that considers the complex interplay between sports, industrial restructuring, and carbon emissions. The universality of the study lies in its applicability to other regions and countries facing similar challenges in promoting sustainable development. The value of the study is that it provides evidence-based policy recommendations that can guide decision-makers in designing effective policies that promote sustainable development through sports.

This paper explores the impact of sports fiscal expenditure on regional carbon emissions and its mechanism of action, which to some extent enriches the study of the carbon reduction effect of government sports fiscal expenditure. However, we do not explore this issue by using prefecture-level cities or smaller administrative units as research objects because of data availability. In the future, we will try to combine the data related to listed companies in the sports industry and match the data at the prefecture-level city level, so as to conduct a more in-depth exploration. In addition, we will further promote the further integration of sports sociology with industrial economics and regional economics in our subsequent research to further enrich the development of sports humanities and social sciences.

**Author Contributions:** Conceptualization, M.X. and Y.X.; methodology, D.W.; software, D.W.; resources, Y.X.; data curation, Y.X.; writing—original draft, M.X. and D.W.; writing—review and editing, M.X. and D.W.; supervision, Y.X.; project administration, M.X.; funding acquisition, Y.X. All authors have read and agreed to the published version of the manuscript.

**Funding:** This research received no external funding.

**Data Availability Statement:** The data used to support the findings of this study are included within the article.

**Conflicts of Interest:** The authors declare that they have no conflict of interest.

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
