# Peer review of "Fiscal Expenditure on Sports and Regional Carbon Emissions: Evidence from China"

_sustainability, doi:10.3390/su15097595_

Round 1

Reviewer 1 Report

Dear Sir

Thank you for inviting me to review this paper. After reviewing carefully this paper, I think that the quality of this paper is acceptable. The authors show the suitable methodology. Some findings can be associated with the objectives. Overall, the paper can be considered for publication.

Before doing this, I have some comments for the authors and I hope that the comments can enhance the quality of the paper.

1/ There are few kinds of font, size of font in the manuscript. Please unify all of them.

2/ The abstract should be improved, especially showing the methodology in this.

3/ The paper should rarely use the footnote. Please cite directly in the manuscript.

4/ But fewer studies have explored the impact of government public expenditure on regional low-carbon development from a sports economy perspective. Showing the few studies as you indicated.

5/ I think that the introduction should show the main findings as well as the main techniques/methodology and the structure of the paper.

6/ The Theoretical analysis and research hypothesis are insufficient. I think that the paper should be further developed by adding some recent studies. How the covid pandemic can impact?

7/ The equation of the model should be cited by previous studies.

8/ The paper should show the Graph for hypothesis 1 and 2, and  mediating variable. This step can strongly support for readers

9/ Page 5 does not show the unit of all variables.

10/ The note *, **, *** for Table 4, 6 should be added.

11/ The test of endogeneity. Which variables can meet endogeneity problems? Please show and list all variables.

12/ The limitations of this study

13/ Please consider https://www.mdpi.com/2071-1050/13/16/8748

Thank you

Author Response

1/ There are few kinds of font, size of font in the manuscript. Please unify all of them. 

Thank you for your comments. We have standardized the font and font size of the manuscript.

2/ The abstract should be improved, especially showing the methodology in this. 

Thank you for your comment. We have added the corresponding content to the abstract as you suggested.

“Abstract: Due to the problems of economic structure and the ways of industrial development many countries have accumulated many ecological problems in the process of economic development, especially the increase in carbon emissions, the greenhouse effect, and the emergence of a series of problems, which makes the global ecosystem suffer severe challenges. Achieving green sustainable development has become a strategic development arrangement for all countries, and as the sports economy is closely linked to regional green development, it has become an important channel to influence regional green development by promoting sports development. Based on theoretical analysis, this paper empirically analyzes and tests the impact of government fiscal support for sports on regional carbon emissions and its mechanisms based on provincial panel data using OLS regression, mediated effects model and threshold model. We argue that the development of green and healthy sports can have a significant impact on regional green development and that government financial support for sports plays an important role in this process. We found that government financial support for sports can significantly reduce regional carbon emissions and thus promote regional green development. From the perspective of mechanism analysis, the government's fiscal support for sports leads to the orderly development of the sports industry and its associated industries, which improves the rationalization of the regional industrial structure and thus reduces regional carbon emissions. Moreover, as the carbon productivity of the region continues to rise, the impact of financial support for sports on regional low-carbon development gradually deepens. This paper confirms the important role of the government's emphasis on sports development in the process of achieving carbon peaking and carbon neutrality, verifies the mediating effect of industrial structure rationalization, and further quantifies the correlation through the threshold effect, extending the study of the influencing factors affecting regional carbon emissions.”

3/ The paper should rarely use the footnote. Please cite directly in the manuscript. 

Thank you for your comments. As per your suggestion, we have removed the footnotes and have included direct quotes in the manuscript.

4/ But fewer studies have explored the impact of government public expenditure on regional low-carbon development from a sports economy perspective. Showing the few studies as you indicated. 

Thank you for your reminder that the presentation here is inaccurate, and we have rewritten the third page of the manuscript for the section.

On the other hand, some scholars have explored the low-carbon development path of the sports industry itself (Hu and Yang,2020; Gong and Gui, 2021; Zhang and Wu, 2022), and the impact of the sports economy on regional low-carbon development (Yang and Liu, 2010), but none of them have considered the effect of governmental actions. Obviously, in promoting regional economic development, the government plays an important role in regional economic and social development through public finance spending (Liu, 2012; Wang et al., 2022), but there is a lack of studies exploring the effect of government announcement spending on regional low-carbon development from the perspective of sports economy.

References:

1.Hu, D., & Yang, Y.. The development of marine sports tourism industry based on low-carbon economy. Journal of Coastal Research. 2020, 112(SI), 97-99.

2.Gong, F., & Gui, Y.. Research on the role of sports industry in economic development based on an ecological perspective. Fresenius Environmental Bulletin. 2021,30(3), 2710-2715.

3.Zhang, X., & Wu, Y.. The Green and Low-Carbon Development Effect of Comprehensive Sports Events: A Quasi-Natural Experiment From China. Front. Environ. Sci. 2022,10, 946993.

4.Yiqun, Y., & Jian, L.. Research on promotion of regional low-carbon economy development by sports tourism industry of city circle. In Proceedings of the 7th International Conference on Innovation & Management. 2010, 315-320.

5.Liu, J. Y.. Analysis of the synergistic impacts of public services expenditure on economic growth and social equity. 2012 International Conference on Management Science & Engineering 19th Annual Conference Proceedings. IEEE, 2012, 1904-1910.

6.Wang, D., Zhang, E., & Liao, H. (2022). Does fiscal decentralization affect regional high-quality development by changing peoples’ livelihood expenditure preferences: Provincial evidence from China. Land. 2022, 11(9), 1407.

5/ I think that the introduction should show the main findings as well as the main techniques/methodology and the structure of the paper. 

Thank you for your comments. We have added the relevant content in the last part of the introduction.

“Based on the theoretical analysis, this paper conducts an empirical analysis based on the collected panel data and uses OLS regression, mediated effects analysis and threshold regression. The paper found that government fiscal support on sports could significantly reduce the level of regional carbon emissions and thus promote regional green development. From the perspective of mechanism analysis, the government's fiscal support on sports could lead to the orderly development of sports industry and its related industries, which improves the rationalization of regional industrial structure and thus reduces the level of regional carbon emissions. In addition, with the continuous improvement of regional carbon productivity, the influence of fiscal support for sports undertakings on regional low-carbon development gradually deepens. The remainder of the paper is designed as follows: the second part is the theoretical analysis and hypotheses, where we provide a logical analysis of the paper's research themes and mechanisms. The third part is the research design, where we introduce the analytical model involved in this paper and explain the relevant variables. The fourth part is the empirical analysis and testing. The fifth part is the conclusion and implication.”

6/ The Theoretical analysis and research hypothesis are insufficient. I think that the paper should be further developed by adding some recent studies. How the covid pandemic can impact?

Thank you for your comments. On the one hand, we add to the theoretical analysis and research hypotheses of the manuscript. On the other hand, our manuscript does not address the impact of the covid pandemic, mainly due to the difficulty in collecting data on government financial expenditures on sports, and although we missed the analysis of the green effect of COVID-19 on government financial expenditures on sports, we will certainly study the relevant issues within this period in the near future when the time is ripe.

7/ The equation of the model should be cited by previous studies. 

Thank you for your reminder. We have cited from previous studies during the use of the model, in p5-p6 of the manuscript.

8/ The paper should show the Graph for hypothesis 1 and 2, and  mediating variable. This step can strongly support for readers.  

Thank you for your comments. We will assume that 1 and 2 and the mediating mechanism have been shown through the Graph, as shown in p6.

Figure 1: Transmission mechanism

9/ Page 5 does not show the unit of all variables.  

Thank you for your comments. We add the units of all variables to the descriptive statistics on page 7.

10/ The note *, **, *** for Table 4, 6 should be added. 

Thank you for your comments. We have added notes to the relevant information in Table 4 and Table 6.

11/ The test of endogeneity. Which variables can meet endogeneity problems? Please show and list all variables. 

Thank you for your comments. In order to eliminate the effect of endogeneity, we adopted the instrumental variables approach to address the endogeneity problem as shown on page 9. In addition, we take variance inflation factor (VIF) test for each variable in the model, and the variance inflation factor (VIF) of each variable is less than 10, and there is no multicollinearity problem.

Variable

VIF

gdp

4.61

consump

3.80

patent

3.07

fdi

2.74

sports

1.58

r&d

1.41

12/ The limitations of this study 

Thank you for your comments. We have added the limitations of our study as well as future perspectives. As shown p 12.

“And this paper explores the impact of sports fiscal expenditure on regional carbon emissions and its mechanism of action, which to some extent enriches the study of the carbon reduction effect of government sports fiscal expenditure. However, we do not explore this issue by using prefecture-level cities or smaller administrative units as research objects because of data availability. In the future, we will try to combine the data related to listed companies in the sports industry and match the data at the prefecture-level city level, so as to conduct a more in-depth exploration. In addition, we will further promote the further integration of sports sociology with industrial economics and regional economics in our subsequent research to further enrich the development of sports humanities and social sciences.”

13/ Please consider https://www.mdpi.com/2071-1050/13/16/8748 

Thank you for your comments. This paper provides a good value for our research and we cite it.

Reviewer 2 Report

This paper aims to study the impacts of government financial support for sports on regional carbon emissions. This paper tested three hypotheses by using econometric analysis of provincial panel data from 2006-2017 in China. This is a good paper offering interesting ideas which merit publication and might be of great interest to many readers of various disciplines. However, this paper is not ready for publication since it still suffers from serious flaws, particularly from the econometric model in the baseline regression which was poorly designed. As a result, the conclusion of the paper is somewhat doubtful and contradicts the existing literature. My comments regarding the model are as follows:

1.      The authors use both fiscal expenditures on sports and GDP in their model. Based on the regression results, the authors show that fiscal expenditure on sports is beneficial for regional emissions reduction, while high GDP is associated with high regional emissions. However, if we refer to the existing literature on the nexus between fiscal expenditures and GDP, those results were somewhat confusing since increasing fiscal expenditures will lead to increasing GDP and vice versa. Hence, increasing fiscal expenditure on sports will likely decrease regional emissions, but at the same time, it will increase regional GDP which will cause more emissions. The same question is also applied to industrial structure rationalization. Have the authors considered the impacts of industrial structure rationalization on GDP?

2.      The authors assume that GDP has a linear impact on regional emissions. Moreover, based on the regression results, the impact of GDP on regional emissions is positive. What are the proper policy recommendations for this finding? Should the government stop economic development to reduce regional emissions? Additionally, by assuming a linear relationship between GDP and emissions, the notion of green development could not be captured by the model. In this regard, I would like to suggest the authors introduce the quadratic terms of GDP in their model.

3.      The coefficient for energy was found to be negative and statistically significant. This variable portrays the level of energy consumption which according to the manuscript was defined as energy consumption per unit of GDP. The negative coefficient of energy implies that a higher level of energy consumption (per unit GDP) leads to a lower level of regional emissions. Firstly, this finding is difficult to accept and contradicts the existing literature on the energy-emission nexus. Secondly, the author’s interpretation of this result cannot be accepted since it did not reflect the finding of the models. For instance, in line 323, the authors argued that: "a reduction in energy consumption per unit of GDP can curb the increase in carbon emissions and promote regional low carbon development". Additionally, in line 420, the authors argued that: "on the one hand, the government should actively promote the green concept deep into the growth and development of sports to activate the vitality of regional green innovation, enhance regional ecological and environmental awareness, reduce energy consumption levels and promote regional green development through green sports"

Minor comments:

The authors should justify why they are using a Tobit model, at least by using a graph to show that your data is indeed left- or right-censored.

Author Response

Thank you for your affirmation of the research topic of our manuscript. The issues you raised have been thoroughly considered and discussed, especially the issue of the selection of control variables and the choice of analytical models.

First, we choose regional GDP per capita as a control variable mainly because it is a non-negligible control variable. We believe that there is a strong relationship between regional economic development and green development. Secondly, the regression coefficient of GDP on carbon emissions is significantly positive, which may be a problem in the model design. Although the diversified regional consumption and the use of energy-intensive equipment may increase carbon emissions as the level of regional development increases, the regional development approach as well as the consumption approach will change toward sustainable development as the regional development reaches a certain level. According to this logical analysis, as you have pointed out and suggested, the impact of GDP on regional carbon emissions may be non-linear. Therefore, we follow your suggestion and we add both the primary and secondary terms of GDP per capita to the model, and the regression results are consistent with our expectation, showing a non-linear relationship.

Second, based on your suggestion, we further discussed the relationship between the level of energy consumption and carbon emissions and the relationship between the level of energy consumption per unit of GDP and other control variables, and we believe that the level of energy consumption (energy) should be excluded from the model we designed. We take variance inflation factor (VIF) test for each variable in the model, and the variance inflation factor (VIF) of each variable is less than 10, and there is no multicollinearity problem. Therefore, the selection of control variables in the modified version of our manuscript is statistically justified. In addition, we provide a brief explanation of the selection of control variables. First, the regional R&D investment can change the level of regional green innovation, which in turn may affect the level of regional carbon emissions (Yang et al., 2022), therefore, we introduce R&D support (r&d) and innovation base and climate (patent) as control variables in the model. Second, some studies also suggest that trade openness may contribute to the development of the renewable energy industry (Nguyen & Nguyen, 2021), and the introduction of external capital may also promote the development of the energy industry and thus affect the level of regional carbon emissions, so we introduce the level of external openness (fdi) as a control variable into the model. Third, we consider gross regional product (gdp) per capita as a non-negligible control variable. We argue that as the income level of regional residents increases, the diversified consumption of residents and the use of high energy-consuming equipment increase regional carbon emissions. Of course, as the income level of regional residents and technological progress continue to increase, residents' consumption may be tilted toward green development. Therefore, the effect of regional gross domestic product (gdp) per capita on regional carbon emissions may be nonlinear, and we add the primary and secondary terms of this variable to the model equation. Fourth, regional consumption capacity may also influence the level of regional green development (Wang et al.,2022), therefore, we introduce the level of resident consumption (consump) into the model.

Variable

VIF

gdp

4.61

consump

3.80

patent

3.07

fdi

2.74

sports

1.58

r&d

1.41

References:

Wang D, Zhang E, Qiu P and Hong X.. Does increasing public expenditure on sports promote regional sustainable development: Evidence from China. Frontiers in Public Health. 2022, 10:976188.

Yang S, Feng D, Lu J, et al. The effect of venture capital on green innovation: Is environmental regulation an institutional guarantee?. Journal of Environmental Management, 2022, 318: 115641.

Nguyen, T.T.; Nguyen, V.C. Financial Development and Renewables in Southeast Asian Countries—The Role of Organic Waste Materials. Sustainability. 2021, 13, 8748.

Minor comments:

The authors should justify why they are using a Tobit model, at least by using a graph to show that your data is indeed left- or right-censored.

Thank you for your comments. As shown by the histogram of the dependent variable and the kernel density function curves, there is a left-censored in the data of the dependent variable. Therefore, we chose to use the Tobit model for testing.

Reviewer 3 Report

The idea of the study is interesting, however, I would like to make several recommendations that would potentially help the authors to improve the quality and readability of the manuscript.

Introduction. What exactly is the research question(s) the authors hope to address? I understand that the paper aims to fill a gap “by estimating the relation between fiscal expenditure on sports and regional carbon emissions”? I am not sure how you arrived at this “gap”, and why the reader should care? The authors need to provide a much more convincing argument for why the government should enhance fiscal expenditure on sports to reduce carbon emissions.   The authors need more space to describe the potential links that exist between the fiscal expenditure on sports and regional carbon emissions, and why this link is worth studying?

Please, extend the literature review to justify the methods of the Interactions between the fiscal expenditure on sports and regional carbon emissions.  The literature review is based basically on an investigation by the authors from China. The problem should be studied globally. It will help the authors highlight controversial and diverging investigation hypotheses. The authors should mention the significance of this research in a few lines.

The research methods are appropriate, but the methodology raises questions. I do not see a justification for the chosen variables. Though few of them are justified in the background analysis, some appear ‘out of nowhere”, e.g. R&D support, innovation base and climate (patent), level of openness to the outside world, level of energy consumption, and per capita gross regional product.

Discussion section needs to be a coherent and cohesive set of arguments that take us beyond this study in particular and help us see the relevance of what the authors have proposed. The author needs to contextualize the literature findings and be explicit about the added value of your study towards that literature. Also, other studies should be cited to increase the theoretical background of each method used. Findings should be contextualized in the literature and explicit about the added value of the study towards the literature.

Please add citations on data sources: China Sports Statistical Yearbook, China Energy Statistical Yearbook, National Bureau of Statistics

Author Response

Comments and Suggestions for Authors

The idea of the study is interesting, however, I would like to make several recommendations that would potentially help the authors to improve the quality and readability of the manuscript.

Thank you for your affirmation of the research theme of our manuscript, and we have discussed the suggestions you gave in a profound way. Once again, thank you for taking the time to guide our manuscript.

1.Introduction. What exactly is the research question(s) the authors hope to address? I understand that the paper aims to fill a gap “by estimating the relation between fiscal expenditure on sports and regional carbon emissions”? I am not sure how you arrived at this “gap”, and why the reader should care? The authors need to provide a much more convincing argument for why the government should enhance fiscal expenditure on sports to reduce carbon emissions.   The authors need more space to describe the potential links that exist between the fiscal expenditure on sports and regional carbon emissions, and why this link is worth studying?

Thank you for your comments. We have revised and improved the content of the introduction section, as shown on pages 2-4.

2.Please, extend the literature review to justify the methods of the Interactions between the fiscal expenditure on sports and regional carbon emissions.  The literature review is based basically on an investigation by the authors from China. The problem should be studied globally. It will help the authors highlight controversial and diverging investigation hypotheses. The authors should mention the significance of this research in a few lines.

Thank you for your comments. Your suggestion we think is very mature, however, there are few studies on the carbon effects of financial spending on sports and we have added to the global studies as appropriate to the actual situation, as shown on p.2-3 of our manuscript.

3.The research methods are appropriate, but the methodology raises questions. I do not see a justification for the chosen variables. Though few of them are justified in the background analysis, some appear ‘out of nowhere”, e.g. R&D support, innovation base and climate (patent), level of openness to the outside world, level of energy consumption, and per capita gross regional product.

Thank you for your comments. We further discussed the relationship between the level of energy consumption and carbon emissions and the relationship between the level of energy consumption per unit of GDP and other control variables, and we believe that the level of energy consumption (energy) should be excluded from the model we designed.

We take variance inflation factor (VIF) test for each variable in the model, and the variance inflation factor (VIF) of each variable is less than 10, and there is no multicollinearity problem. Therefore, the selection of control variables in the modified version of our manuscript is statistically justified.

In addition, we provide a brief explanation of the selection of control variables. First, the regional R&D investment can change the level of regional green innovation, which in turn may affect the level of regional carbon emissions (Yang et al., 2022), therefore, we introduce R&D support (r&d) and innovation base and climate (patent) as control variables in the model. Second, some studies also suggest that trade openness may contribute to the development of the renewable energy industry (Nguyen & Nguyen, 2021), and the introduction of external capital may also promote the development of the energy industry and thus affect the level of regional carbon emissions, so we introduce the level of external openness (fdi) as a control variable into the model. Third, we consider gross regional product (gdp) per capita as a non-negligible control variable. We argue that as the income level of regional residents increases, the diversified consumption of residents and the use of high energy-consuming equipment increase regional carbon emissions. Of course, as the income level of regional residents and technological progress continue to increase, residents' consumption may be tilted toward green development. Therefore, the effect of regional gross domestic product (gdp) per capita on regional carbon emissions may be nonlinear, and we add the primary and secondary terms of this variable to the model equation. Fourth, regional consumption capacity may also influence the level of regional green development (Wang et al.,2022), therefore, we introduce the level of resident consumption (consump) into the model.

4.Discussion section needs to be a coherent and cohesive set of arguments that take us beyond this study in particular and help us see the relevance of what the authors have proposed. The author needs to contextualize the literature findings and be explicit about the added value of your study towards that literature. Also, other studies should be cited to increase the theoretical background of each method used. Findings should be contextualized in the literature and explicit about the added value of the study towards the literature. 

Thank you for your comments. On the one hand, we have improved the Discussion section, as shown p.13. On the other hand, we have added literature support for each model equation (1)-(7) used in the manuscript, as shown p.6.

5.Please add citations on data sources: China Sports Statistical Yearbook, China Energy Statistical Yearbook, National Bureau of Statistics  

Thank you for your comments. We have added links to relevant databases in the manuscript.

Round 2

Reviewer 1 Report

Dear Sir/Madam

After checking the revised paper, I agree with all revisions that were revised by the authors.

I agree that this paper should be considered for publication.

Thank you

Author Response

Thank you for your comments or suggestions during the manuscript revision process, and we look forward to receiving more of your guidance and help in the future.

Reviewer 2 Report

I appreciate the authors’ remarkable effort to revise the manuscript following my comments in the previous round of the review. However, I still find some serious flaws in the manuscript that make the manuscript is not ready for publication.

1.      The authors claimed in their responses to the reviewer’s comments that GDP is a non-negligible control variable for carbon dioxide emissions, so it has to be included in their model. How about energy consumption? Don’t you think that energy consumption is also a non-negligible control variable for carbon dioxide emissions? There is a huge amount of literature showing that energy consumption is a stronger prediction of carbon dioxide emissions compared to GDP. Hence, if you removed energy consumption as a control variable, your model will suffer from omitted variable bias.

2.      The authors claimed that they have removed the level of energy consumption from their model. However, I still found the discussion on energy consumption in section 3.1., i.e., in equation (1) and in the explanation following equation (1).

3.      In the revised manuscript, the authors put their previously published paper as the main reference for their model in equation (1). Your previous model included the degree of population agglomeration in the region as one of the independent variables. Why did you remove it from your current model?

4.      The authors claimed that their dependent variable is left-censored. By doing so, the authors provided a histogram of the dependent variable with its kernel density function curve. However, the histogram did not show the data is left-censored. Instead, the data is either right-skewed or left-truncated. The authors should differentiate between left-censored, left-truncated, and right-skewed data. Hence, the authors still failed to justify the use of Tobit regression.

5.      In the abstract, the authors explained that their model was estimated by using OLS regression. Which one is the correct one? OLS regression or Tobit?

6.      The authors found that the coefficient of the quadratic term of the GDP is negative and statistically significant. After confirming the non-linearity of the GDP, the authors should try to estimate the turning point of the GDP, of which further improvement of GDP will be beneficial for carbon dioxide emission reduction. Along with an explanation of whether this turning point is achievable or not.

7.      In the revised manuscript, the authors introduced a new independent variable that captures the regional consumption capacity which is proxied by the level of residential consumption. By definition, GDP is a sum of consumption, investment, and government spending. What is the justification to include regional consumption in your model? Hasn’t it already been captured by the GDP?

8.      What do the authors mean by making guided and regulated consumption patterns (page 9, line 391)? Can you elaborate more and give a clear example?

9.      The financial support for sports, GDP, and regional consumption are having the same dimension. However, the authors treated those variables differently, i.e., by taking the logarithmic transformation for financial support for sports but taking the per capita level forms for the other two variables. What is the justification for this different treatment?

Author Response

I appreciate the authors’ remarkable effort to revise the manuscript following my comments in the previous round of the review. However, I still find some serious flaws in the manuscript that make the manuscript is not ready for publication.

Thank you for your comments. Your comments and suggestions are a key factor in improving the quality of our manuscripts.

1.The authors claimed in their responses to the reviewer’s comments that GDP is a non-negligible control variable for carbon dioxide emissions, so it has to be included in their model. How about energy consumption? Don’t you think that energy consumption is also a non-negligible control variable for carbon dioxide emissions? There is a huge amount of literature showing that energy consumption is a stronger prediction of carbon dioxide emissions compared to GDP. Hence, if you removed energy consumption as a control variable, your model will suffer from omitted variable bias.

Thank you for your comments, we have removed this control variable after careful consideration. We agree that your level of energy consumption has a significant impact on carbon emissions. And we have focused on carbon productivity as a threshold variable in our threshold effects analysis. On the one hand, the threshold model analysis leads to the important conclusion that the carbon-reducing effect of sports fiscal spending, for example, is more significant as regional carbon productivity increases, which emphasizes the importance of carbon productivity. On the other hand, the analysis with the addition of threshold variables reveals that the conclusion of the carbon reduction effect of sports fiscal spending does not change, and such an approach may further confirm the robustness of the analysis results. Therefore, although we exclude the level of energy consumption from the control variables, we compensate and strengthen the interpretation of this issue in the subsequent analysis.

2.The authors claimed that they have removed the level of energy consumption from their model. However, I still found the discussion on energy consumption in section 3.1., i.e., in equation (1) and in the explanation following equation (1).

Thank you for your comments. We overlooked this work in our revision process, and thanks to your reminder, we removed the content.

3.In the revised manuscript, the authors put their previously published paper as the main reference for their model in equation (1). Your previous model included the degree of population agglomeration in the region as one of the independent variables. Why did you remove it from your current model?

Thank you for your comments. The dependent variables in our manuscript as well as the units of the control variables control for the factor of population size and take a per capita variable for measurement. In this case, the effect of population size and agglomeration on regional per capita carbon emissions is not significant. Therefore, we did not include this control variable in the current model.

4.The authors claimed that their dependent variable is left-censored. By doing so, the authors provided a histogram of the dependent variable with its kernel density function curve. However, the histogram did not show the data is left-censored. Instead, the data is either right-skewed or left-truncated. The authors should differentiate between left-censored, left-truncated, and right-skewed data. Hence, the authors still failed to justify the use of Tobit regression.

Thank you for your guidance and assistance. The dependent variable of the manuscript is unilaterally constrained and truncated to the left; therefore, we adopted multiple models, including the Tobit model, for regression analysis in the baseline regression analysis.

5.In the abstract, the authors explained that their model was estimated by using OLS regression. Which one is the correct one? OLS regression or Tobit?

Thank you for your comments. The regression analysis in our article is based on OLS regression, and the use of Tobit model regression is in the baseline regression to serve as a multiple model analysis to test the robustness of the results. Thanks to your reminding, we have revised the expression of the abstract section.

6.The authors found that the coefficient of the quadratic term of the GDP is negative and statistically significant. After confirming the non-linearity of the GDP, the authors should try to estimate the turning point of the GDP, of which further improvement of GDP will be beneficial for carbon dioxide emission reduction. Along with an explanation of whether this turning point is achievable or not.

Thank you for your comments. According to the results of the baseline regression in Table 2 of this paper and the regression analysis after eliminating the endogeneity problem in Table 3, the inflection point of GDP per capita obtained in our model is around 110900 RMB (as shown p.9). Of course this result is after avoiding the endogeneity problem, and if necessary, we will focus on the topic of the relationship between GDP per capita and regional carbon emissions in our subsequent study.

7.In the revised manuscript, the authors introduced a new independent variable that captures the regional consumption capacity which is proxied by the level of residential consumption. By definition, GDP is a sum of consumption, investment, and government spending. What is the justification to include regional consumption in your model? Hasn’t it already been captured by the GDP?

Thank you for your comments. Based on your suggestions and tips, we have removed the control variable of residential consumption level after careful discussion.

8.What do the authors mean by making guided and regulated consumption patterns (page 9, line 391)? Can you elaborate more and give a clear example?

Thank you for your comments. We have removed the control variable of the level of consumption of the population. Therefore, we have also removed the relevant elements of the interpretation of the regression results.

  1. The financial support for sports, GDP, and regional consumption are having the same dimension. However, the authors treated those variables differently, i.e., by taking the logarithmic transformation for financial support for sports but taking the per capita level forms for the other two variables. What is the justification for this different treatment?

Thank you for your comment. Sports infrastructure built by fiscal expenditure on sports is a public good or public service, which is theoretically non-exclusive and non-competitive. Therefore, the economic and social effects generated by public fiscal expenditure on sports should be regionally holistic. However, the level of regional economic development is more commonly measured by GDP per capita rather than gross value, which better reflects this meaning and corresponds better to the dependent variable.

Reviewer 3 Report

Thank you for the revised paper. The authors have corrected all mentioned issues

Author Response

Thank you for your comments or suggestions during the manuscript revision process, and I look forward to receiving more of your guidance and help in the future.

Round 3

Reviewer 2 Report

After the second round of the review, I did not find any significant improvement in the manuscript. The manuscript still suffers from serious flaws and is not ready for publication. The authors still failed to address the fundamental issues regarding their model and data. Moreover, the emission models became worse since the authors decided to exclude the most influential predictor of emission in their model. The followings are my detailed comments:

1.      Previously, I raised concerns about the problems with the omitted variable bias. The authors disregarded this issue and argued that they did that on purpose after taking careful consideration. Additionally, the authors argued that their subsequent analysis will compensate for this serious flaw.

My response:

I would like to remind the author that in the presence of omitted-variable bias, the OLS estimator will be biased and inconsistent. The problems occurred not only in your baseline models but also in your subsequent models since your subsequent models contain controls from the baseline model. Hence, instead of compensating for the flaws in your baseline models, your subsequent models will carry over the flaws in the baseline model. As a result, all of your interpretations of the model should be rejected and all of your hypotheses cannot be verified.

Allow me to discuss it from a different perspective. Let’s assume that I did not know that the authors removed the variable of energy consumption from their models simply because it had an unexpectedly negative and significant coefficient. In the manuscript, the authors argued that their baseline model was based on their previously published paper. In that paper, the authors argued that energy consumption had a significant impact on green and sustainable development. Hence, simply by referring to your previously published paper, your so-called careful consideration to exclude energy consumption from the emission model cannot be accepted.

2.      Previously, I asked for justification for the use of the Tobit regression. In the second round of the review, the authors admitted that their data is left-truncated, that’s why they were using the Tobit regression.

My response:

Since you have already admitted that your data is left-truncated, then instead of using Tobit regression, you should use truncated regression. Again, the authors still failed to justify the use of Tobit regression.

3.      Previously, I questioned the purpose of taking the logarithmic transformation for the variable of financial support for sports. However, I did not get a satisfactory answer from the authors. In some cases, the logarithmic transformation was taken to remove the skewness of the original data. Do you have a similar problem with the variable of financial support for sports, or do you have any other problems? What about the data on emissions and GDP? Why did not you take logarithmic transformations for those variables? It is very uncommon to have a level-log regression with emissions and GDP in level, but financial support is in the log form.

Author Response

After the second round of the review, I did not find any significant improvement in the manuscript. The manuscript still suffers from serious flaws and is not ready for publication. The authors still failed to address the fundamental issues regarding their model and data. Moreover, the emission models became worse since the authors decided to exclude the most influential predictor of emission in their model. The followings are my detailed comments:

Thank you for your reviews of the model and data section of our manuscript. We have given careful consideration to the questions you have raised, and we may have misunderstood and confused you by not answering the first two rounds of questions adequately or by not considering the issues comprehensively enough.

1.Previously, I raised concerns about the problems with the omitted variable bias. The authors disregarded this issue and argued that they did that on purpose after taking careful consideration. Additionally, the authors argued that their subsequent analysis will compensate for this serious flaw.

My response:

I would like to remind the author that in the presence of omitted-variable bias, the OLS estimator will be biased and inconsistent. The problems occurred not only in your baseline models but also in your subsequent models since your subsequent models contain controls from the baseline model. Hence, instead of compensating for the flaws in your baseline models, your subsequent models will carry over the flaws in the baseline model. As a result, all of your interpretations of the model should be rejected and all of your hypotheses cannot be verified.

Allow me to discuss it from a different perspective. Let’s assume that I did not know that the authors removed the variable of energy consumption from their models simply because it had an unexpectedly negative and significant coefficient. In the manuscript, the authors argued that their baseline model was based on their previously published paper. In that paper, the authors argued that energy consumption had a significant impact on green and sustainable development. Hence, simply by referring to your previously published paper, your so-called careful consideration to exclude energy consumption from the emission model cannot be accepted.

Thank you for your suggestions on problems related to control variables for energy consumption levels.

In the first two rounds we removed the indicator of energy consumption level considering the issue of threshold effects analysis in the latter part of the article, adding here whether there is redundancy. This issue, as you pointed out, should avoid or reduce the problem of model omitted variables in the first place. Therefore, we added the level of energy consumption as a control variable in the current round of revisions. It can be seen that the regression coefficient of energy consumption level on regional carbon emissions is significantly negative. The regression coefficients of regional energy consumption levels (energy) on regional carbon emissions are all negative and pass the significance test at the 1% confidence interval. This result may seem surprised, but in practical terms such an empirical analysis is reasonable. The decreasing amount of energy consumption per unit of GDP due to technological progress has been accompanied by increasing living standards and consumption levels of the population, which has increased per capita carbon emissions.

2.Previously, I asked for justification for the use of the Tobit regression. In the second round of the review, the authors admitted that their data is left-truncated, that’s why they were using the Tobit regression.

My response:

Since you have already admitted that your data is left-truncated, then instead of using Tobit regression, you should use truncated regression. Again, the authors still failed to justify the use of Tobit regression.

Thank you for your suggestions regarding the use of the model.

In the first two rounds of responses you questioned our language and lack of clarity about the solution to the specific problem. In this paper, we add a truncated regression model to the baseline regression analysis. Furthermore, to check the robustness of the manuscript regression results, we used multiple regression models for the regression analysis, including truncated regression model, random effects model, and Tobit model, based on the fixed effects regression analysis. The final regression coefficients of sports expenditure on regional carbon emissions are all significantly negative at 1% confidence interval, which indicates that our regression results are significant. Thank you for your suggestions on the use of the models.

3.Previously, I questioned the purpose of taking the logarithmic transformation for the variable of financial support for sports. However, I did not get a satisfactory answer from the authors. In some cases, the logarithmic transformation was taken to remove the skewness of the original data. Do you have a similar problem with the variable of financial support for sports, or do you have any other problems? What about the data on emissions and GDP? Why did not you take logarithmic transformations for those variables? It is very uncommon to have a level-log regression with emissions and GDP in level, but financial support is in the log form.

Thank you for your comments.

Our previous answers were not comprehensive and clear enough to cause your confusion.

We adopt a logarithmic approach to sports fiscal expenditure for the following two main considerations. First, from an economic perspective, sports infrastructure and public goods and services are public goods or public services (Ren and Chen, 2016) with non-exclusive and non-competitive properties, and sports infrastructure construction may have obvious externalities and social effects. In addition, in China's development history, regional sports finances are not exactly spent according to population demand. Second, from the perspective of data, since there are extreme values or large differences in sports expenditure data, in order to reduce the absolute differences between data and avoid the influence of individual extreme values, we adopt a logarithmic treatment for sports expenditure data, which can reduce the multicollinearity and eliminate the influence of the magnitude to a certain extent, thus satisfying the linear model assumption as much as possible. Third, carbon emissions and regional development levels are usually measured in per capita form in the existing literature, while the public attribute of sports expenditure is measured in non-per capita form, which may reflect the positive externality of sports expenditure (Feng and Jia, 2018).

References:

[1]Ren Changsheng, Chen Linhui. Mass sport Public Goods Supply in a Balancing Urban and Rural Development Perspective [J].Sichuan Sports Science ,2016,35(04):91-94+115. DOI:10.13932/j.cnki.sctykx.2016.04.23.

[2]Feng Guoyou, Jia Shanghui.Research on the Commitment, Practice and Effect of China’s Fiscal Policy Support for Sport Industry Development[J].China Sport Science,2018,38(09):37-46. DOI:10.16469/j.css.201809004.

Thank you for taking the time to make reasonable suggestions for our manuscript. We hope our revisions will meet the standards and wish you a happy life and good work! We have benefited from your guidance and look forward to working with you!

Round 4

Reviewer 2 Report

After the third round of the review, despite the authors’ efforts to revise the manuscript, I still found serious flaws in the manuscript which made the manuscript not ready for publication. The followings are my detailed comments:

1.      The authors have included energy consumption back into their emission model, but the authors still failed to make a proper interpretation of the regression results. I agree with the authors that in some cases, the sign of a certain coefficient might not be as we expected. However, the authors should be able to explain this unexpected result clearly to the readers. Regarding the unexpectedly negative sign of energy consumption, the authors argued that:

This result may seem surprised, but in practical terms such an empirical analysis is reasonable. The decreasing amount of energy consumption per unit of GDP due to technological progress has been accompanied by increasing living standards and consumption levels of the population, which has increased per capita carbon emissions.

My response:

The increasing living standards and consumption levels of the population have already been captured by GDP, which according to your model, the impact of GDP on emissions followed the inverted U-shaped relationship. This implies that the increasing living standards and consumption levels of the population do not always lead to increasing per capita carbon emissions. Hence, the authors’ argument was unreasonable and refuted by their model.

My suggestion:

Instead of making an unreasonable argument, the authors should try to explain this unexpected result by using a statistical approach. For instance, the authors might consider building nested models to make comparisons between the full model and the reduced model. By doing so, the authors might be able to find the suppressor effects (if any) and explain why the energy consumption had an unexpectedly negative coefficient.

2.      The authors have included truncated regression in their model due to the distribution of the dependent variable which is left-truncated.  

My response:

§  The newly added paragraph (lines 364-373) contains too many technical and grammatical errors.

§  What is the meaning of the following statement (lines 367-368): To test the robustness of the regression results, we adopted a regression analysis by replacing the regression model.

§  If you have confirmed that your data is left-truncated and you have used the truncated regression, why did you still include the Tobit regression in your analysis?

§  Again, if you have confirmed that your data is left-truncated, why did you use the fixed effect regression for calculating the turning point of the GDP? Why did not you use the truncated regression as your final model?

3.      Again, I did not get a satisfactory answer from the authors regarding their approach to taking the logarithmic transformation for the financial support for sports variable.

My response:

§  The authors claim that they use the logarithmic transformation to deal with extreme values or large differences in data. Please kindly support your argument with a reliable reference from a reputable statistics, econometrics, or mathematics journal/book, not from non-Scopus/WoS sports journals that were published in the Chinese language.

§  Why did you prefer to use the logarithmic transformation compared to the “per capita transformation” as you did with GDP and emissions? Please kindly support your argument with a reliable reference from a reputable statistics, econometrics, or mathematics journal/book, not from non-Scopus/WoS sports journals that were published in the Chinese language.

Author Response

After three rounds of responses and four challenges from Reviewer 2, our manuscript collaborators responded as follows.

First, the subject of this paper is the effect of sports fiscal expenditure on regional carbon emissions, and the reviewer2 has always only questioned the control variables selected in the model construction of the article, which has no impact on the quality of the manuscript. In his opinion, he believes that the control variables are the ones that should be the main explanatory variables of the manuscript, including GDP per capita, energy consumption levels, etc. When we cited published literature to support our selected control variables and adjusted the control variables as he suggested, he illogically compared the regression coefficients of the control variables in the references with the control variables in this paper on the basis of ignoring the inconsistency of the explanatory variables and completely rejected all the work of our manuscript. In the actual regression analysis, the effect of each variable should exist periodically, and according to the logic of the reviewer it can be seen that the model selection of the articles he has published using linear models are all wrong, and this biased perspective on the issue makes us question this.

Second, on the basis of his selection of the review option of not being qualified to comment on the language, it is unfounded that after three rounds of revisions, which we extremely patiently made according to reviewer 2, he suggested that there were problems with the grammar of the article. In addition, we could see in multiple rounds of communication that this reviewer's language skills were lacking and the language was basically translated by the AI. Even then we revised the language of our manuscript again, and this was done only to improve our manuscript.

Third, Reviewer 2 consistently questioned the model selection of our manuscript, and after we added truncated regressions as he suggested, he continued to find some implausible issues. Our manuscript adopts fixed effects as the main effect in model selection, as most manuscripts studying development economics issues do, and adopts multiple regressions for robustness testing so as to ensure the validity of the regression conclusions. However, the reviewer disregarded the overall control of the article and his comments were not scientific and objective, especially after the second round directly pointed out that the literature we referenced was our previously published article.

Fourth, we answer in great detail the question of adopting a logarithmic treatment for profound sports economic expenditures from an economic perspective and a statistical perspective. Moreover, we justified our adoption of this treatment by considering the sports economy topic we studied and the fact that there are few studies in this area, and we supported it with similar treatments in two journals, Sichuan Sports Science and China Sport Science. The basic response of reviewer 2 showed subjective dissatisfaction twice in a row, and he also argued that the fact that these two journals are Chinese journals cannot be used as a basis. The message we want to convey to you is that these two journals are the top authoritative Chinese journals in the field of Kinesiology sponsored by the authority official.In addition, we can probably tell if the reviewer2, who is also Chinese, has violated the basic professional qualities of a journal reviewer with such an opinion and distorted style of work in his heart.

Therefore, due to the unprofessional and subjective review of this reviewer, we are here to make a preliminary reflection of this issue to your journal and expect to be dealt with, and we reserve all legal avenues to appeal against the actions of this reviewer. We hope that the relevant workers in the editorial board will handle our issue correctly.
